



# Improving Latin American Soil Information Database for Digital Soil Mapping enhances its usability and scalability

Sergio Diaz-Guadarrama[1], Iván Lizarazo[1] Mario Guevara[2-4], Marcos Angelini[5], Gustavo A. Araujo-Carrillo[6], Jainer Argeñal[7], Daphne Armas[8], Rafael A. Balta[9], Adriana Bolivar[10], Nelson Bustamante[11], Ricardo O. Dart[12], Martin Dell Acqua[13], Arnulfo Encina[14], Hernán Figueredo[15], Fernando Fontes[13], Joan S. Gutierrez-Diaz[16], Wilmer Jiménez[17], Raúl S. Lavado[18], Jesús F Mansilla-Baca[12], Maria de Lourdes Mendonça-Santos[12], Lucas M. Moretti[19], Iván D. Muñoz[20], Carolina Olivera[5], Guillermo Olmedo[5], Christian Omuto[5], Sol Ortiz[21], Carla Pascale[22], Marco Pfeiffer[23], Iván A. Ramos[24], Danny Ríos[25], Rafael Rivera[26], Lady M. Rodriguez[20], Darío M. Rodríguez[27], Albán Rosales[28], Kenset Rosales[29], Guillermo Schulz[27], Víctor Sevilla[30], Leonardo M. Tenti[27], Ronald Vargas[5], Viviana M. Varón-Ramírez[6], Gustavo M. Vasques[12], Yusuf Yigini[5], Yolanda Rubiano[1].

[1]Departamento de Agronomía, Facultad de Ciencias Agrarias. Universidad Nacional de Colombia, Bogotá, Colombia
[2]Centro de Geociencias - Universidad Nacional Autónoma de México Campus Juriquilla, Querétaro, 76230, México.
[3]University of California, Riverside, Department of Environmental Sciences, Riverside CA. 92507, USA.
[4]United States Department of Agriculture, Soil Salinity National Laboratory, Riverside CA. 92507, USA.
[5]FAO, Vialle de Terme di Caracalla, Rome, Italy
[6]Corporación Colombiana de Investigación Agropecuaria AGROSAVIA, C.I. Tibaitatá, Bogotá, CO-0571, Colombia
[7]Facultad de Ciencias/ Universidad Nacional Autónoma de Honduras, Honduras.
[8]Departamento de Agronomía, Edif. CITEIIB. Universidad de Almería. Almería, 04120, España
[9]Dirección General de Asuntos Ambientales Agrarios, Ministerio de Desarrollo Agrario y Riego, Perú
[10]Subdirección Agrología, Instituto Geográfico Agustín Codazzi, Bogotá, Colombia
[11]Servicio Agrícola y Ganadero, Santiago de Chile, Chile
[12]Embrapa Solos, Rio de Janeiro, 22460-000, Brasil.
[13]Direccion General de Recursos Naturales, Ministerio de Ganadería, Agricultura y Pesca, Montevideo, Uruguay
[14]Facultad de Ciencias Agrarias de la Universidad Nacional de Asunción, Asunción, Paraguay
[15]Sociedad Boliviana de la Ciencia del Suelo, La Paz, Bolivia.
[16]Department of Agroecology, Faculty of Science and Technology, Aarhus University, Tjele, DK-8830 Denmark
[17]Ministerio de Agricultura y Ganadería, Quito, 170516, Ecuador.
[18]Facultad de Agronomía e INBA (CONICET/UBA), Universidad de Buenos Aires, Buenos Aires, 1417, Argentina.
[19]Estación Experimental Agropecuaria Cerro Azul, Instituto Nacional de Tecnología Agropecuaria, Misiones, Argentina.
[20]Subdirección de Geografía, Instituto Geográfico Agustín Codazzi - IGAC, Bogotá, 111321, Colombia
[21]Secretaría de Agricultura y Desarrollo Rural, México.
[22]Ministerio de Agricultura, Ganadería y Pesca (MAGYP), Argentina
[23]Departamento de Ingeniería y Suelos, Facultad de Ciencias Agronómicas, Universidad de Chile, Santiago, Chile.
[24]Instituto de Investigación Agropecuaria de Panamá, Ciudad de Panamá, Panamá
[25]Departamento de Ciencias del Suelo y Ordenamiento Territorial, Universidad Nacional de Asunción, Paraguay.
[26]Ministerio de Medio Ambiente, Santo Domingo, República Dominicana
[27]Instituto de Suelos (CIRN), Instituto Nacional de Tecnología Agropecuaria, Hurlingham, Buenos Aires, B1686, Argentina.
[28]Instituto de Innovación en Transferencia y Tecnología Agropecuaria, San José, Costa Rica
[29]Ministerio de Ambiente y Recursos Naturales, Guatemala.
[30]Universidad Central de Venezuela, Maracay, Venezuela.

*Correspondence to*: Sergio Díaz (sediazg@unal.edu.co), Mario Guevara (mguevara@geociencias.unam.mx)



**Abstract.** Spatial soil databases can help model complex phenomena in which soils are decisive, for example, evaluating agricultural potential or estimating carbon storage capacity. The Soil Information System for Latin America and the Caribbean, SISLAC, is a regional initiative promoted by the FAO's South American Soil Partnership to contribute to the sustainable management of soil. SISLAC includes data coming from 49,084 soil profiles distributed unevenly across the continent, making it the region's largest soil database. However, some problems hinder its usages, such as the quality of the data and its high dimensionality. The objective of this research is twofold. First, to evaluate the quality of SISLAC and its data values and generate a new, improved version that meets the minimum quality requirements to be used by different interests or practical applications. Second, to demonstrate the potential of improved soil profile databases to generate more accurate information on soil properties, by conducting a case study to estimate the spatial variability of the percentage of soil organic carbon using 192 profiles in a 1473 km$^2$ region located in the department of Valle del Cauca, Colombia. The findings show that 15 percent of the existing soil profiles had an inaccurate description of the diagnostic horizons. Further correction of an 4.5 additional percent of existing inconsistencies improved overall data quality. The improved database consists of 41,691 profiles and is available for public use at https://doi.org/10.5281/zenodo.6540710 (Díaz-Guadarrama, S. & Guevara, M., 2022). The updated profiles were segmented using algorithms for quantitative pedology to estimate the spatial variability. We generated segments one centimeter thick along with each soil profile data, then the values of these segments were adjusted using a spline-type function to enhance vertical continuity and reliability. Vertical variability was estimated up to 150 cm in-depth, while ordinary kriging predicts horizontal variability at three depth intervals, 0 to 5, 5 to 15, and 15 to 30 cm, at 250 m-spatial resolution, following the standards of the GlobalSoilMap project. Finally, the leave-one-out cross-validation provides information for evaluating the kriging model performance, obtaining values for the RMSE index between 1.77% and 1.79% and the R$^2$ index greater than 0.5. The results show the usability of SISLAC database to generate spatial information on soil properties and suggest further efforts to collect a more significant amount of data to guide sustainable soil management.

## 1 Introduction

Soil is a three-dimensional natural body consisting of strata called horizons when there are chemical, biological, and even physical relations (i.e., transference of components or products of their alteration among them) or simply layers when they are a consequence of successive deposition of different sediments. Bot, horizons, and layers are a mixture of degraded mineral materials, organic material, air, and water (Bockheim et al., 2005). Soil is a product of the soil itself (such a point information on a site), climate, organisms, topography, parent material, time, and spatial position, also known as the SCORPAN factors of soil formation (Mcbratney et al., 2003). The soil provides various ecologic or productive contributions besides the obvious importance as a critical factor in food production, e. g. in urban ecosystem services (such a water buffering capacity of open areas), human health (breakdown of toxic contaminants), or climate regulation through carbon storage (Otte et al., 2012). Its sustainable management is of the utmost importance in the main environmental challenges





such as food security, climate change, and the loss of biodiversity (Dewitte et al., 2013). Soil data are an essential starting point to reach an adequate level of knowledge about soil status, raise awareness about its importance and preserve this valuable resource (Bouma et al., 2012). Digital soil data (such as soil profiles) are in great demand as inputs to, for example, estimate the potential of agricultural land (Amirinejad et al., 2011; Bini et al., 2013; Owusu et al., 2020); in addition, their

availability is key to assess soil functions such as water and climate regulation, energy supply and biodiversity (Greiner et al., 2017). Greater dissemination of soil information has substantial benefits in disciplines such as agricultural sciences by allowing better estimation of current and future crop productivity or identifying constraints and risks of land degradation (FAO & IIASA, 2009; Hopmans et al., 2021; Paterson et al., 2015). FAO indicates that more and better soil data can drive achievements in the fight against poverty and hunger as well as to advance sustainable development(FAO, 2017).

Technological advances and increased computing capabilities have led to the development of soil databases at regional and global scales (Hendriks et al., 2019; Keskin et al., 2019; Rossiter, 2018). Global databases such as the World Soil Information Service, WoSIS (Batjes et al., 2017, 2020), or World Inventory of Soil Property Estimates, WISE (Batjes, 2016), regional databases such as Soil Profiles in Africa (Leenaars, 2013), as well as national ones such as SISINTA in Argentina (Angelini et al., 2018), or IRAKA in Colombia (Araujo-Carrillo et al., 2021) exist. These datasets are an example

of efforts at different levels to have soil profile data that helps to support decision-making in problems involving this resource's management. Organizations such as FAO, the Global Soil Partnership (GSP), and the Latin America and the Caribbean Soil Partnership (LACS), emphasize the need to preserve such data due as, in some parts of the world,  soil survey data are the only source of information available (Beaudette & O'Geen, 2009; Hengl & Macmillan, 2019).

The mentioned above databases allow scientists to generate information on soil properties and estimate soil organic carbon

(SOC). SOC is one of the most important chemical properties related to soil fertility and climate regulation, the key to multiple functions in ecosystem services (Owusu et al., 2020). Global projects such as the FAO Organic Carbon Map (FAO & ITPS, 2018), national projects in Brazil (Gomes et al., 2019), Ghana (Owusu et al., 2020), Cameroon (Silatsa et al., 2020) or regional projects in Andalusia, Spain (Armas et al., 2017), or in paramo ecosystem soils in Colombia (Gutierrez et al., 2020); have been some of the works that have estimated SOC (in its vertical or horizontal dimensions) from soil databases.

Soil Information System for Latin America and the Caribbean, SISLAC, is an initiative coordinated and financed by the FAO Global Soil Partnership to contribute to the sustainable management of this resource in the region (SISLAC, 2013). SISLAC (Fig. 1a) has data on almost 50,000 soil profiles and 140,000 horizons and layers, making it the most extensive database in the region. The data includes a description of the site for each profile, its spatial location, the layers that comprise it, its physical and chemical properties, data provider, and metadata. However, when analyzing the SISLAC data, it is

evident that some of them present inconsistencies due to the high heterogeneity of sources that provide such data. These inconsistencies can be due to, for example, old descriptions using obsolete description systems or errors in transcriptions from field to office. So, if they are not corrected, the analysis results will have a high degree of uncertainty and inaccuracies, primarily since the performance of a model depends on the quality of the training data (Garg et al., 2020).

Data quality is a multidimensional concept involving management, analysis, quality control, storage, and presentation
(Chapman, 2005). It is closely related to their potential use and ability to meet user needs (English, 1999), which Krol
(2008) calls "use aptitude".

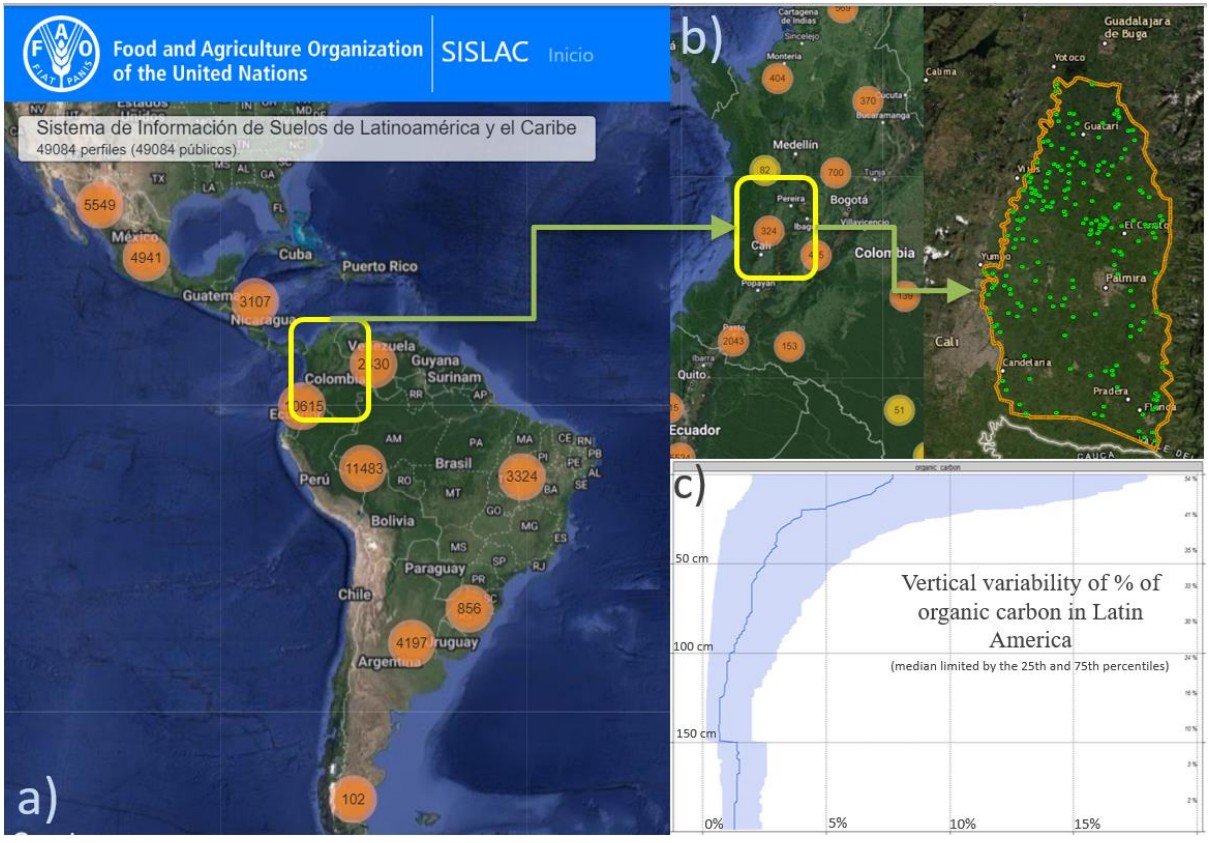

**Figure 1, a) SISLAC interface, each number in the orange circles indicates the number of profiles in that area (from SISLAC
webpage); b) Location of the data usability demonstration area (ESRI 2022); c) Vertical variability of the percentage of organic**
**carbon in Latin America.**

Therefore, this research aims to: (i) evaluate the quality of the SISLAC data in terms of logical consistency; (ii) improve the
quality of the data to provide a new updated version; and (iii) demonstrate the usability, applicability, and potential of
SISLAC to support digital soil mapping and soil-related policy research in South America by assessing the vertical and
horizontal variability of SOC percentage (as in Fig. 1c) in a region of Valle del Cauca Colombia. Two factors were
considered for selecting the case study zone: (i) to be an area of agricultural production; and (ii) to have a relatively high
density of soil profiles with SOC values.





## 2 Data and Methods

The flow diagram (Fig. 2) shows the work carried out, consisting of two phases. The first phase comprises processes of validation and debugging of errors and inconsistencies in the SISLAC data. The second phase focuses on analysis of the usability demonstration using the spatial variability of the SOC in a specific site.

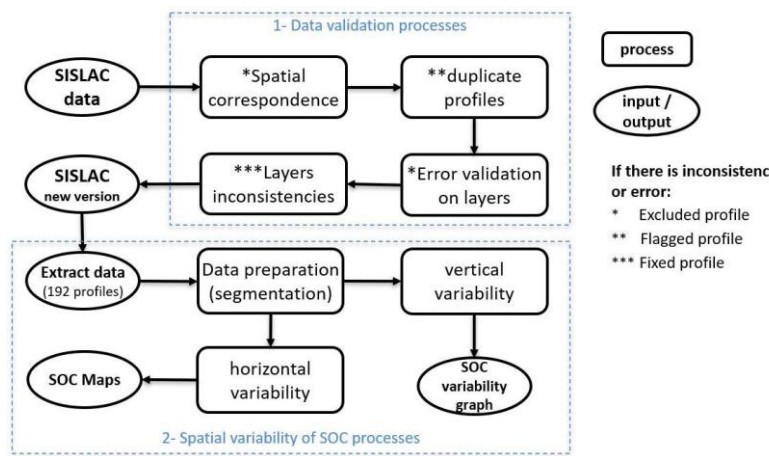

**Figure 2, Flowchart of this research. The first part (upper frame) consists of verification of spatial correspondence, profile duplication, debugging of errors and inconsistencies. The second part (lower frame) is about data preparation and estimation of the spatial variability of the SOC.**

### 2.1 Study area

The study area (Fig. 1a) is composed of the Latin American and Caribbean countries listed in Table 1, where since 2016 we have a soil database representative of such a diverse region. In the same figure, the number of profiles per region can be seen aggregated in orange circles. In addition, an agricultural area located in the department of Valle del Cauca, Colombia (Fig 2a), was selected as case study zone to demonstrate usability. This area is located between latitudes 3°15' and 3°51' N and longitudes 75°57' and 76°10' W. The altitude of the area varies between 900 and 1,000 meters above sea level, and it has an approximate area of 1,437 square kilometers.

### 2.2 Data

The SISLAC database, which can be downloaded from the official site (http://54.229.242.119/sislac/es), consists of 49,084 profiles (with a total of 139,746 horizons). The number of these by country is detailed in Table 1. For the first part of this research, 100% of the data were analyzed, while for the analysis of the spatial variability of the SOC, 192 profiles corresponding to the case study zone were used and their distribution is shown in Fig. 1b.



**Table 1, Initial profiles and their layers by country. The countries are ordered by number of profiles, those with less than 100 profiles were grouped together. NA: Not Applicable**

| Country | Profiles | Layers |
|---|---|---|
| **Ecuador** | 13056 | 36749 |
| **México** | 12223 | 26051 |
| **Brazil** | 7842 | 23926 |
| **Colombia** | 4864 | 18900 |
| **Argentina** | 3774 | 16902 |
| **Paraguay** | 2830 | 6041 |
| **Bolivia** | 2557 | 2773 |
| **Venezuela** | 1056 | 4108 |
| **Uruguay** | 272 | 1382 |
| **Peru** | 148 | 631 |
| **Jamaica, Costa Rica, Cuba.** | Between 100 and 51 | NA |
| **Chile, Guyana, Puerto Rico, Surinam, Nicaragua.** | Between 50 and 26 | NA |
| **Panamá, Guatemala, Belice, Honduras, El Salvador, French Guiana, The Antilles, Barbados, Virgin islands, Trinidad y Tobago, República Dominicana.** | Less than 26 | NA |
| **Total** | 49084 | 139746 |

Profile attributes are detailed in Table 2, in this the name of the attribute is listed in the first column, description in the second and data type in the third. The location is given in geographic coordinates, WGS84 datum. While for horizons and layers, their attributes are listed in Table 3 in the same way as in the profiles.

**Table 2. Profiles attributes, attributes related to the site description.**

| Column name | Description | Type |
|---|---|---|
| **profile_identifier** | Profile identifier | text |
| **latitude** | Profile latitude. Decimal degrees | numeric |
| **longitude** | Profile longitude. Decimal degrees | numeric |
| **country_code** | Country code. ISO 3166-1 | text |
| **date** | Survey date | YYYY-MM-DD |
| **source** | data source | text |
| **contact** | Contact e-mail about the data | text |
| **order** | Soil order | text |
| **type** | Type (profile, auger) | text |
| **license** | License code (PDDL, ODC-By, ODC-ODbL, CC-BY, CC-BY-NC, CC-BY-NC-ND) | text |



**Table 3. Layers attributes, the measured attributes are numerical attributes (excluding top and bottom, which are the limits of each layer), in the last column, for each attribute measured, the percentage of records with valid data is indicated. NA: Not applicable**

| Column name | Description | Units | % of layers with data |
|---|---|---|---|
| **profile_identifier** | Profile identifier | text | NA |
| **layer_identifier** | Unique ID of each horizon | text | NA |
| **designation** | Layer nomenclature | text | NA |
| **top** | Upper limit | numeric | NA |
| **bottom** | Lower limit | numeric | NA |
| **bulk_density** | Bulk density | numeric | 15.2 |
| **ca_co3** | Inorganic carbon (%) | numeric | 5.7 |
| **coarse_fragments** | Coarse fragments (%) | numeric | 5.3 |
| **ecec** | Effective cation exchange capacity | numeric | 39.5 |
| **conductivity** | Electric conductivity | numeric | 23.6 |
| **organic_carbon** | Organic carbon (%) | numeric | 57.1 |
| **ph** | pH specified with metadata | numeric | 75.8 |
| **clay** | Clay (%) | numeric | 75.2 |
| **silt** | Silt (%) | numeric | 59.7 |
| **sand** | Sand (%) | numeric | 73.5 |
| **water_retention** | Water retention (%) | numeric | 3.1 |

**2.3 Methods**

**2.3.1 Quality assessment and improvement of SISLAC data**

The evaluation of the quality and improvement of the SISLAC data were carried out in parallel in three stages, the first two for the site data and the third for the different layers. The first stage consisted of checking that the profiles are in the correct location (spatial correspondence). It was carried out by spatial intersection between the profiles (points) and the cartography of the countries (polygons). Based on the *country_code* attribute of the profiles, this correspondence was verified, those that coincided with their respective country were considered valid (Fig. 3a). Those that did not coincide were verified one by one,

those that were within the limits of their country, considering the cartographic scale of the reference information, the precision of the equipment with which the coordinate was taken, or the reference systems under which original data were taken, they were considered valid (Fig. 3b). Still, others had the coordinates inverted (Fig. 3c), the latitude and longitude values were exchanged, and their correspondence was verified again. Finally, the profiles outside their zone that could not be corrected for having the wrong location were excluded (Fig. 3d).





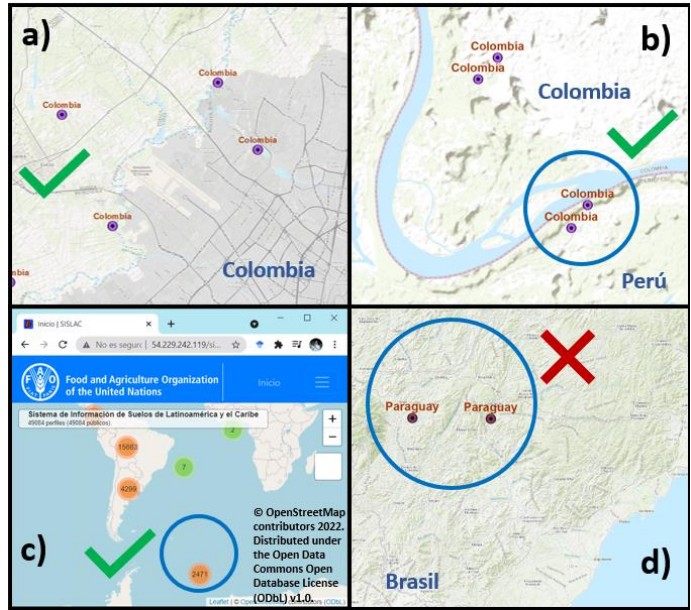

**Figure 3, Example of criteria found in spatial validation, (figures a, b and d source ESRI 2022; c: SISLAC webpage)**

The second stage consisted of verifying if there are overlapping profiles, in addition, to verifying if the values in their attributes are different. For this, the number of times the same pair of coordinates is repeated was massively validated. Unlike the previous validation, these cannot be arbitrarily excluded since the correct profile cannot be determined. Then, those with duplicity were marked, so the user of the data can use the ones he considers appropriate. A new attribute in the profiles (*perfil_duplicado* of binary type) indicates if the profile has duplicity (TRUE) or is unique (FALSE).

The third stage consisted of validating the description of the horizons or layers of each profile, verifying: $u_1 < v_1 \leq u_2 < v_2 \leq \ldots \leq u_n < v_n$; where $u$ is the upper limit and $v$ the lower limit. The upper limit must be less than its lower limit, and the lower limit must be less than or equal to the upper limit of the next layer. Gaps may exist but never overlap between layers. Errors were first validated, those in which the structure could not be corrected, so the profiles were excluded. Table 4 lists the three applied rules, their description, and an example of these.





**Table 4, Layer errors validation. In the example, the layers with errors are highlighted in bold letters, for the first and third case, the last layers of the profiles are the ones with error, while in the second case, both layers have error because the limits have no data.**

| Validation | Description | Example |
|---|---|---|
| Duplicated layers | Layer limits are duplicated, and the values of the attributes are different. | **ID Perfil · ID Horizonte · Top · Bottom · SOC %**<br>176583 · 846371 · 0 · 10 · 32.4<br>176583 · 846371 · 10 · 23 · 26.1<br>**176583 · 846371 · 23 · 30 · 27.3**<br>**176583 · 846371 · 23 · 30 · 2.1** |
| Empty limits | Upper and lower limits do not contain data. | **ID Perfil · ID Horizonte · Top · Bottom · SOC %**<br>Santa Rosa · Santa Rosa-1 · · · 1.22<br>Santa Rosa · Santa Rosa-2 · · · 0.68 |
| Layers overlap | Layers overlap in a profile. | **ID Perfil · ID Horizonte · Top · Bottom · SOC %**<br>SD-107050 · SD-107050-1 · 0 · 5 · 1.14<br>SD-107050 · SD-107050-2 · 5 · 20 · 0<br>SD-107050 · SD-107050-3 · 20 · 60 · 0.43<br>**SD-107050 · SD-107050-4 · 60 · 90 · 0**<br>**SD-107050 · SD-107050-5 · 40 · 130 · 0**<br>**SD-107050 · SD-107050-6 · 130 · 150 · 0** |

After excluding the profiles with errors, the existence of inconsistencies was validated. Unlike errors, these can be corrected by guidelines that do not alter the structure of the profile. Next, Table 5 lists the rules applied to their description and the guideline for their correction. For a better understanding of the content of Table 5, Table 6 below illustrates the described inconsistency (middle column) and how it was corrected (third column).

**Table 5, Description of the validation of inconsistencies and their correction guideline.**

| Validation | Description | Correction Guideline |
|---|---|---|
| Organic layer | When the first layer is described in the opposite direction and from the second the normal description begins. Layer commonly known as organic. | Invert the values of the first layer and rescale subsequent limits based on the thickness of the organic layer. |
| Inverted layer | The value of the limits of a layer is inverted, it is verified considering also the previous and later layers. | Invert the values of the layer. |
| Continuous final layer | The value of the lower limit of the last layer is empty | Assign the value of the upper limit of the last layer plus 10. |
| duplicated layer | Horizon that presents duplicate layers in all its attributes. | Delete duplicated layers. |
| Upper limit is null | The upper limit of a layer is null, in addition, the lower limit of that layer and the previous one is not null. | Assign the lower limit value of the previous layer. |
| Lower limit is null | The lower limit of a layer is null, in addition, the upper limit of that layer and the next are not null. The last layer is not validated. | Assign the value of the upper limit of the next layer. |





**Table 6, Illustration of inconsistencies and their correction guideline. In the second column in bold type the layers with inconsistency are shown, in the third column also in bold type it is shown how to correct them using the established guidelines. In the first case all profile limits are modified, for the rest only those of the layer with inconsistency.**

| Validation | Inconsistency | Correction Guideline |
|---|---|---|
| Organic layer | **ID Perfil / ID Horizonte / Top / Bottom / SOC %**<br>**C-03 / C-03-1 / 5 / 0 / **<br>**C-03 / C-03-2 / 0 / 5 / 3.9**<br>C-03 / C-03-3 / 5 / 25 / 1.1<br>C-03 / C-03-4 / 25 / 40 / 0.7<br>C-03 / C-03-5 / 40 / 77 / 0.3<br>C-03 / C-03-6 / 77 / 115 / 0.3<br>C-03 / C-03-7 / 115 / 180 / 0.2 | **ID Perfil / ID Horizonte / Top / Bottom / SOC %**<br>C-03 / C-03-1 / **5** / **0** /<br>C-03 / C-03-2 / **0** / **5** / 3.9<br>C-03 / C-03-3 / **5** / **25** / 1.1<br>C-03 / C-03-4 / **25** / **40** / 0.7<br>C-03 / C-03-5 / **40** / **77** / 0.3<br>C-03 / C-03-6 / **77** / **115** / 0.3<br>C-03 / C-03-7 / **115** / **180** / 0.2 |
| Inverted layer | **ID Perfil / ID Horizonte / Top / Bottom / SOC %**<br>**ICAG-TOL-22 / ICAG-TOL-22-1 / 7 / 0 /**<br>ICAG-TOL-22 / ICAG-TOL-22-2 / 7 / 21 / 9.48<br>ICAG-TOL-22 / ICAG-TOL-22-3 / 21 / 45 / 4.72<br>ICAG-TOL-22 / ICAG-TOL-22-4 / 45 / 87 / 1.09<br>ICAG-TOL-22 / ICAG-TOL-22-5 / 87 / 120 / 1.1<br>ICAG-TOL-22 / ICAG-TOL-22-6 / 120 / 170 / 1.02 | **ID Perfil / ID Horizonte / Top / Bottom / SOC %**<br>**ICAG-TOL-22 / ICAG-TOL-22-1 / 0 / 7 /**<br>ICAG-TOL-22 / ICAG-TOL-22-2 / 7 / 21 / 9.48<br>ICAG-TOL-22 / ICAG-TOL-22-3 / 21 / 45 / 4.72<br>ICAG-TOL-22 / ICAG-TOL-22-4 / 45 / 87 / 1.09<br>ICAG-TOL-22 / ICAG-TOL-22-5 / 87 / 120 / 1.1<br>ICAG-TOL-22 / ICAG-TOL-22-6 / 120 / 170 / 1.02 |
| Continuous final layer | **ID Perfil / ID Horizonte / Top / Bottom / SOC %**<br>ICAG-TOL-35 / ICAG-TOL-35-1 / 0 / 12 / 0.76<br>ICAG-TOL-35 / ICAG-TOL-35-2 / 12 / 64 / 0.21<br>ICAG-TOL-35 / ICAG-TOL-35-3 / 64 / 85 / 0.1<br>ICAG-TOL-35 / ICAG-TOL-35-4 / 85 / 140 / 0.1<br>**ICAG-TOL-35 / ICAG-TOL-35-5 / 140 / / 0.1** | **ID Perfil / ID Horizonte / Top / Bottom / SOC %**<br>ICAG-TOL-35 / ICAG-TOL-35-1 / 0 / 12 / 0.76<br>ICAG-TOL-35 / ICAG-TOL-35-2 / 12 / 64 / 0.21<br>ICAG-TOL-35 / ICAG-TOL-35-3 / 64 / 85 / 0.1<br>ICAG-TOL-35 / ICAG-TOL-35-4 / 85 / 140 / 0.1<br>**ICAG-TOL-35 / ICAG-TOL-35-5 / 140 / 150 / 0.1** |
| Duplicated layer | **ID Perfil / ID Horizonte / Top / Bottom / SOC %**<br>176583 / 846371 / 0 / 10 / 32.4<br>176583 / 846372 / 10 / 23 / 26.1<br>**176583 / 846373 / 23 / 30 / 27.3**<br>**176583 / 846374 / 23 / 30 / 27.3** | **ID Perfil / ID Horizonte / Top / Bottom / SOC %**<br>176583 / 846371 / 0 / 10 / 32.4<br>176583 / 846372 / 10 / 23 / 26.1<br>**176583 / 846373 / 23 / 30 / 27.3** |
| Upper limit is null | **ID Perfil / ID Horizonte / Top / Bottom / SOC %**<br>ICAG-VAC-C1 / ICAG-VAC-C1-H1 / 0 / 12 / 8.52<br>ICAG-VAC-C1 / ICAG-VAC-C1-H2 / 12 / 38 / 2.66<br>ICAG-VAC-C1 / ICAG-VAC-C1-H3 / 38 / 68 / 1.06<br>**ICAG-VAC-C1 / ICAG-VAC-C1-H4 / / 90 / 0.84**<br>ICAG-VAC-C1 / ICAG-VAC-C1-H5 / 90 / 150 / 0.55 | **ID Perfil / ID Horizonte / Top / Bottom / SOC %**<br>ICAG-VAC-C1 / ICAG-VAC-C1-H1 / 0 / 12 / 8.52<br>ICAG-VAC-C1 / ICAG-VAC-C1-H2 / 12 / 38 / 2.66<br>ICAG-VAC-C1 / ICAG-VAC-C1-H3 / 38 / 68 / 1.06<br>**ICAG-VAC-C1 / ICAG-VAC-C1-H4 / 68 / 90 / 0.84**<br>ICAG-VAC-C1 / ICAG-VAC-C1-H5 / 90 / 150 / 0.55 |
| Lower limit is null | **ID Perfil / ID Horizonte / Top / Bottom / SOC %**<br>**Perfil 48081 / 0 / 0 / / 4.72**<br>**Perfil 48081 / 18 / 18 / / 1.09**<br>**Perfil 48081 / 37 / 37 / / 1.1**<br>**Perfil 48081 / 70 / 70 / / 1.02** | **ID Perfil / ID Horizonte / Top / Bottom / SOC %**<br>**Perfil 48081 / 0 / 0 / 18 / 4.72**<br>**Perfil 48081 / 18 / 18 / 37 / 1.09**<br>**Perfil 48081 / 37 / 37 / 70 / 1.1**<br>**Perfil 48081 / 70 / 70 / / 1.02** |

After applying the above validations, a new harmonized database for Latin America is obtained from soil profiles that have minimum integrity requirements. The following is an exercise to demonstrate the usability of this database, taking soil organic carbon in percentage as a target variable and digital soil mapping as a practical approach.

**2.3.2 Data Usability**

195    As mentioned in the introduction, the case study zone was selected for its availability of profiles, however, this exercise can be replicated by applying small changes to the code, which is available as part of this work. It should be considered that the



chosen area should preferably be homogeneous and have a good density of profiles. The above is intended to demonstrate the potential of this database.

With the 192 profiles corresponding to the case study zone, the vertical and horizontal variability of the SOC was estimated. For the latter, the spatial resolution was 250 meters at three depth intervals: 0 to 5, 5 to 15 and 15 to 30 cm, following the standards of the project GlobalSoilMap (2015). As a first step, to harmonize the profiles —using the R software (R Core Team, 2018)— these were segmented using the *slice* function of the *aqp* library (Beaudette et al., 2013), which generates so many one-centimeter segments thick as the maximum depth of each profile. However, the values for each segment are inherited from the corresponding horizon, which generates a discontinuous or staggered representation that does not correspond to reality (Malone et al., 2017). To make their values more representative, they were adjusted using the equal area spline proposed by Bishop, et al. (1999) and available (*ea_spline* function) in the *ithir* library (Malone et al., 2009). An example is shown in Fig. 4 of the original profiles (a), their segmentation (b) and their adjusted values (c).

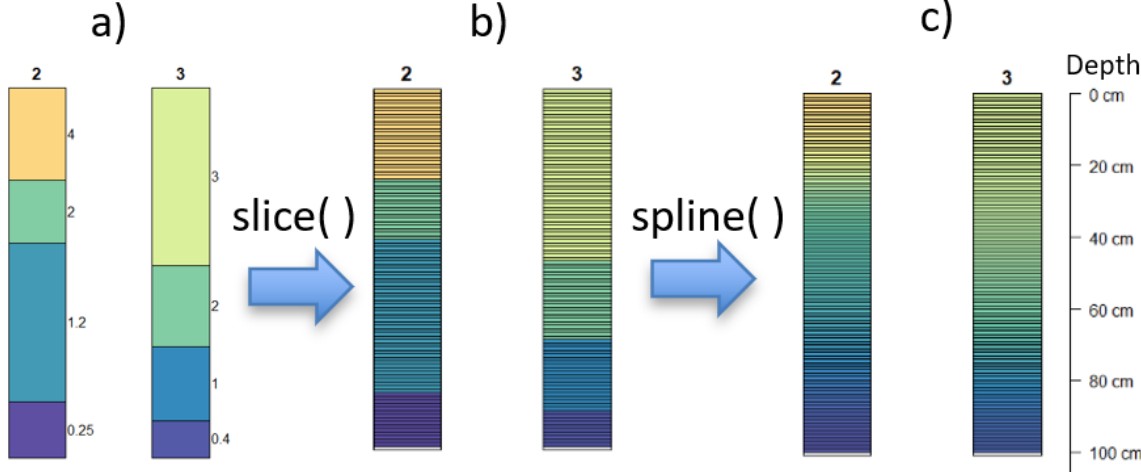

**Figure 4, Harmonization of soil profiles, a) normal representation of the horizons and their SOC percentage; b) segmented horizons, in these the SOC percentage value (the same as the previous one) and c) horizons segmented and with adjusted values to improve their representation using the equal areas spline.**

To calculate vertical variability, the aggregation function of the AQP package was used, which generates statistics for each depth segment (quantiles 5, 25, 50, 75, 95 and percentage of profiles used). From the data generated it is possible to know the behavior of continuous soil characteristics as a function of depth. On the other hand, ordinary kriging (OK) was used for horizontal variability, frequently used to estimate SOC (Bhunia et al., 2018; Duan et al., 2020; Yao et al., 2019; Y. Zhang et al., 2020; Z. Zhang et al., 2020). For each of the three intervals, the SOC percentage value of each profile corresponds to the average of the range of the previously adjusted and segmented values. First, the variogram was generated for each depth and fitted to a theoretical model to obtain the optimal values for interpolation. The estimation of values was carried out and the resulting information was classified according to three categories established by the *Instituto Geográfico Agustín Codazzi* (2016): low: less than 1.2%; medium: between 1.2% and 2.4%; high: greater than 2.4%. Finally, leave-one-out cross-





validation was used for validating the performance of the OK and the root mean squared error (RMSE) and the coefficient of determination ($R^2$) indices were calculated. The Eq. (1) and (2) respectively used for the indices described are the following:

$$RMSE = \left[\frac{1}{n}\sum_{i=1}^{n}(p_i - o_i)^2\right]^{1/2} \tag{1}$$

$$R^2 = 1 - \frac{\sum_{i=1}^{n}(p_i-o_i)^2}{\sum_{i=1}^{n}(o_i-\bar{o}_i)^2} \tag{2}$$

where $o_i$ represents the observed values, $p_i$ the values estimated and $n$ is the number of locations used for the prediction.

### 3 Results

With the first validation, 2726 profiles were found that did not match their country. Table 7 lists these profiles at the country level. As can be seen, Bolivia has the largest number of these with 2,472 (90% of the cases). After the review, it was identified that 2471 of those cases (from Bolivia) had the coordinates inverted, so after changing the values and their

validation, their correct location was verified, and they were considered valid. A total of 36 profiles (1.3% of those reviewed) were excluded for having an erroneous location, as presented in Fig. 3d, 3 from Mexico and 33 from Paraguay. A total of 49,048 profiles (of the initial 49,084) passed the second validation.

**Table 7. Spatial validation results, sorted by country with the highest number of inconsistencies (second column), the third column indicates how many profiles were excluded and the fourth column indicates how many were considered valid after being reviewed**
**one by one.**

| Country | Inconsistent profiles | Excluded profiles | Valid profiles after check |
|---|---|---|---|
| **Bolivia** | 2472 | 0 | 2472 |
| **Colombia** | 78 | 3 | 75 |
| **Paraguay** | 53 | 33 | 20 |
| **Ecuador** | 45 | 0 | 45 |
| **México** | 28 | 0 | 28 |
| **Brazil** | 16 | 0 | 16 |
| **Argentina** | 8 | 0 | 8 |
| **Nicaragua and Venezuela** | 5 | 0 | 5 |
| **Antillas** | 4 | 0 | 4 |
| **Peru and Uruguay** | 3 | 0 | 3 |
| **Chile and Costa Rica** | 2 | 0 | 2 |
| **Vírgen Islands and Jamaica** | 1 | 0 | 1 |
| **Total profiles** | **2726** | **36** | **2690** |

With the second part of the validations, 1989 duplicate profiles were identified. Table 8 lists the country and the number of these. Brazil concentrates the largest amount with 1,680, 84.5% of the total and 21% of the total profiles provided by that





country (with 7,842). As commented in the previous section, the profiles with duplicity were marked in the table, the profiles
with duplicity in the *perfil_duplicado* field contain the value *TRUE*.

**Table 8, Profiles with spatial duplication by country.**

| Country | duplicated profiles |
|---|---|
| Brazil | 1680 |
| Argentina | 94 |
| Colombia | 50 |
| Jamaica | 40 |
| Venezuela | 28 |
| Uruguay | 16 |
| Surinam | 11 |
| Guatemala | 9 |
| Bolivia, Ecuador, Honduras, México | 7 |
| El Salvador, Guyana and Nicaragua. | 6 |
| Panamá | 5 |
| Costa Rica and Peru | 4 |
| Cuba | 2 |
| **TOTAL** | **1989** |

Regarding the revision of the horizons, 7,380 errors were found (in 7,357 profiles). Table 9 details the number of these by country and type. Most were presented in Mexico, Paraguay and Brazil. Profiles with empty limits were the main error with 6,831 cases. Those 7,357 profiles were excluded for being inconsistent.

**Table 9, Layers error validation, the profiles with errors may be fewer than the errors per country because one profile may have more than one type of error.**

| Country | Duplicated layers | Empty limits | Layers overlap | Errors by country | Profiles with error |
|---|---|---|---|---|---|
| **México** | 16 | 4942 | 32 | 4990 | 4990 |
| **Paraguay** | 0 | 1866 | 0 | 1866 | 1866 |
| **Brasil** | 35 | 12 | 339 | 386 | 368 |
| **Colombia** | 1 | 4 | 32 | 37 | 36 |
| **Ecuador** | 0 | 0 | 22 | 22 | 22 |
| **Argentina** | 4 | 2 | 12 | 18 | 18 |
| **Venezuela** | 1 | 4 | 10 | 15 | 13 |
| **Cuba** | 0 | 0 | 12 | 12 | 12 |
| **Costa Rica** | 1 | 0 | 9 | 9 | 8 |
| **Uruguay** | 3 | 0 | 5 | 8 | 7 |
| **Peru** | 0 | 0 | 6 | 6 | 6 |
| **Jamaica** | 0 | 0 | 4 | 4 | 4 |
| **Nicaragua** | 0 | 0 | 4 | 4 | 4 |
| **Chile** | 1 | 1 | 1 | 3 | 3 |
| **Errors by type** | **62** | **6831** | **488** | **7380** | **7357** |

Inconsistencies are described in Table 10. Most were found in Paraguay, Argentina and Colombia. The main causes were the null lower limit, continuous final horizon and duplicate horizon. All of these were corrected according to the established





guidelines. Although 5474 inconsistencies were found, these correspond to 2215 profiles, so there were profiles with more

than one inconsistency, for example, although in Paraguay there are 4066 inconsistencies, these are present in 931 profiles, the same number of profiles in that country.

**Table 10, Layers inconsistencies validation, in these, the bottom limit is null validation was the only one that did not present records with this inconsistency.**

| Country | Organic layer | Inverted layer | Continuous final layer | Duplicated layer | Lower limit is null | Inconsistencies by country. |
|---|---|---|---|---|---|---|
| **Paraguay** | 0 | 0 | 931 | 0 | 3135 | **4066** |
| **Argentina** | 0 | 0 | 993 | 0 | 2 | **995** |
| **Colombia** | 38 | 5 | 0 | 339 | 0 | **382** |
| **Brazil** | 0 | 3 | 0 | 11 | 0 | **14** |
| **Venezuela** | 2 | 0 | 7 | 0 | 0 | **9** |
| **México** | 0 | 1 | 1 | 1 | 0 | **3** |
| **Uruguay** | 0 | 0 | 3 | 0 | 0 | **3** |
| **Bolivia** | 0 | 0 | 1 | 0 | 0 | **1** |
| **Jamaica** | 0 | 0 | 1 | 0 | 0 | **1** |
| **Total by type** | **40** | **9** | **1937** | **351** | **3137** | **5474** |

Finally, Table 11 shows a summary of the data after the validation and correcting processes. Only those countries that had changes due to excluded profiles are listed. The second and third columns show the initial and valid profiles, respectively; the corresponding number of horizons is indicated in parentheses. The Errors column indicates the number of errors in the profiles for that country and inconsistencies is the number of inconsistencies found and corrected. After the processes carried out, of the 49,084 initial profiles, 15% of these were excluded and another 4.5% were corrected so that they met the

minimum integrity requirements. The revised version consists of 41,691 profiles made up of 129,355 horizons and layers.





**Table 11, Details of the SISLAC data validation processes, total number of layers are in parentheses, the errors caused the profile to be excluded, while the inconsistencies were corrected.**

| Country | Initial profiles (layers) | Remain profiles (layers) | Errors | Inconsistencies |
|---|---|---|---|---|
| Ecuador | 13056 (36749) | 13034 (36582) | 22 | 0 |
| México | 12223 (26051) | 7233 (20913) | 4990 | 3 |
| Brazil | 7842 (23926) | 7474 (22616) | 368 | 14 |
| Colombia | 4864 (18900) | 4825 (17615) | 39 | 382 |
| Argentina | 3774 (16902) | 3756 (16813) | 18 | 995 |
| Paraguay | 2830 (6041) | 931 (4066) | 1899 | 4066 |
| Venezuela | 1056 (4108) | 1043 (4051) | 13 | 9 |
| Uruguay | 272 (1382) | 265 (1321) | 7 | 3 |
| Peru | 148 (631) | 142 (561) | 6 | 0 |
| Jamaica | 76 (361) | 72 (331) | 4 | 1 |
| Costa Rica | 55 (318) | 47 (257) | 8 | 0 |
| Cuba | 52 (282) | 40 (186) | 12 | 0 |
| Chile | 45 (220) | 42 (201) | 3 | 0 |
| Nicaragua | 26 (132) | 22 (99) | 4 | 0 |

### 3.2 Data Usability

With the 192 profiles processed which did not present errors or inconsistencies in the validation process, using the aggregation function of the *aqp* library, the SOC vertical variation is shown in Fig. 5, the blue line corresponds to the median, while the shading around it corresponds to at the 25th and 75th percentiles, that is, the variability of 50% of the SOC data. As can be seen, from 0 to 50 cm depth, the median values varies from 1.6% to 0,5%, respectively. While the variability of 50% of the data for the same interval ranges from 0.3% in the minimum values to 2.3% in the maximum values. After 50 270 cm of depth, the values stabilize, with a median value of 0.5% to 0.3% and almost constant variation up to 150 cm.of depth.





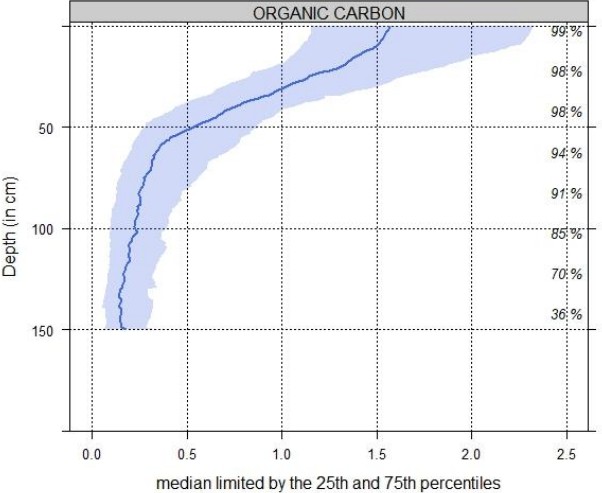

**Figure 5, Vertical variability of the SOC in the area of interest**

Semivariograms obtained allowed us to know the spatial behavior of the profiles. Figure 6 shows that for the first two depths the resulting parameters were similar, while for the third one the range increases and the adjustment model is different. The resulting cartography is shown in Fig. 7, in which it is observed that the estimates have the same distribution patterns of the different categories, although in the third depth (15 to 30 cm) the spot of low category increases. Table 12 shows details of the area percentages for each depth interval and each category. It is observed that the medium category predominates in the three depths mapped with more than 80%, while the low category increases slightly with depth, the inverse being the case in the high category, which decreases.

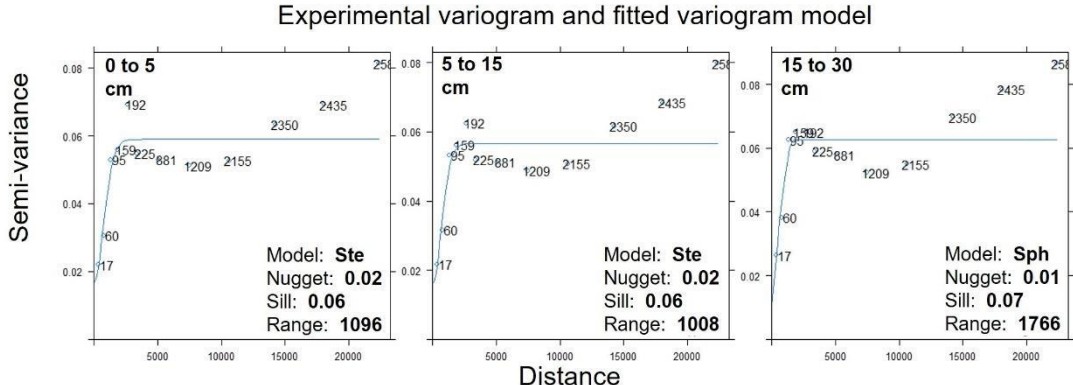

**Figure 6, Adjusted variograms for the three depths, the first two fit the same model (Stein parameterization), with similar range, nugget and sill values, while the third fit a spherical model, its range was considerably larger and the nugget and sill values are similar to the previous ones.**



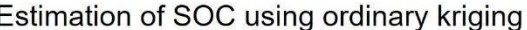

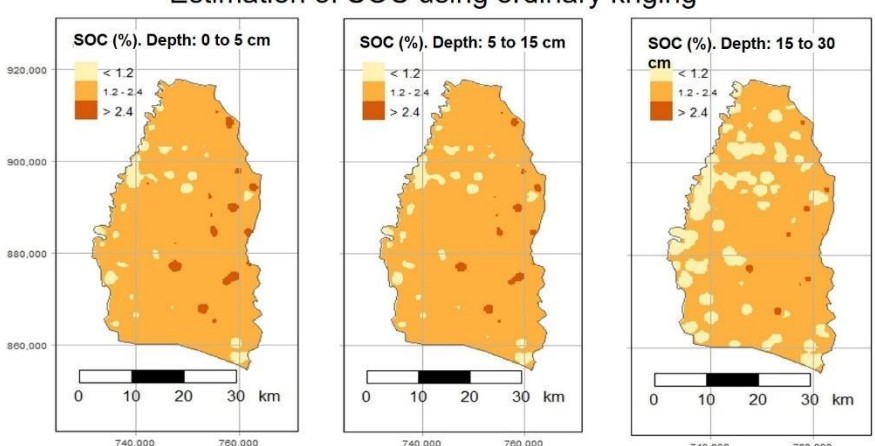

**Figure 7, interpolation results for each depth, orange color predominates, which represents a medium SOC percentage content, as the depth increases the SOC percentage decreases and more yellow patches are observed, mainly in the western zone.**

**Table 12, Percentages of area by depth and category, the values for the 0 to 5 and 5 to 15 cm intervals show very similar percent areas, while the 15 to 30 interval shows what was observed in Fig. 7, that the percent SOC decreases.**

|  | Depth 1: 0- 5 cm | Depth 2: 5 - 15 cm | Depth 3: 15 - 30 cm |
|---|---|---|---|
| **% SOC low** | 5.2 | 5.8 | 19 |
| **% SOC medium** | 92.6 | 92.6 | 80.4 |
| **% SOC high** | 2.2 | 1.6 | 0.6 |

Finally, to evaluate the kriging performance, using leave-one-out cross-validation, the RMSE and $R^2$ indices were obtained. Fig. 8 shows the results of these indexes, as can be seen, the RMSE value was similar for the three intervals, 1.78% from 0 to 5 cm, 1.77% from 5 to 15 cm and 1.79% from 15 to 30 cm. While the resulting $R^2$ was 0.56, 0.53 and 0.83, respectively.



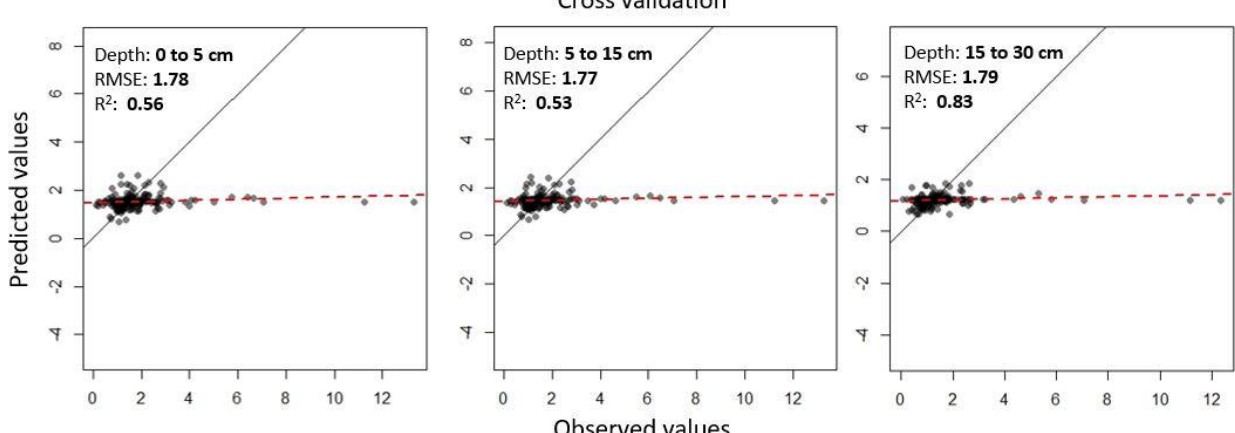

**Figure 8, Cross validation, some subestimated values are observed towards the right side of the graphs, the RMSE values are**
**similar, while the R² for the last interval increases notably.**

## 4 Discussion

This work made it possible to identify that the main problems in the SISLAC profiles occur systematically in some countries. In addition, it shows the potential of improved soil databases for the generation of spatial information such as SOC or any other property which have been surveyed.

As shown in Table 10, the most frequent error in the profiles was due to empty limits, which occur mainly in Mexico and Paraguay with 67% and 25% of the total errors, respectively. In Mexico, these errors correspond to 40% of the profiles provided, while in Paraguay to 65%. On the other hand, most of the inconsistencies (Table 11) are found in Argentina, Paraguay and Colombia with 44%, 42% and 12% of the total respectively. Although all these inconsistencies were corrected, it is observed that, for example, in Paraguay of the total profiles provided (2830), only 9 contain SOC values, the rest have
all the empty attributes. The foregoing represents a limitation if one wanted to carry out any type of analysis with these data. The validations were defined by expert judgment, they coincide with those described in the works of  Batjes (1995) and Leenaars (2013) and were applied to all the elements. For the horizons, it was guaranteed that they were correctly described, since as these authors indicate, if they are not adequately described, in-depth analyzes cannot be carried out since the analysis tools may fail or a high degree of uncertainty may be generated.

The variability allowed knowing the behavior of the SOC in its vertical and horizontal dimensions, the latter following standards for the elaboration of spatial information on soil properties such as those of GlobalSoilMap. An important aspect is that with the segmentation and adjustment of the values carried out, it is possible to generate information for any interval, or even for each centimeter of depth.

This work is a effort towards the consolidation and availability of more and better data in the region, which should be
reflected in national results such as those of Araujo-Carrillo et al. (2021) and Varón-Ramírez et al. (2022) in Colombia;





Armas et al. (2022) in Ecuador; Pfeiffer et al. (2020) in Chile or Schulz et al. (2022) in Argentina. Free access to these data can increase the knowledge of the properties or improve the existing one. It can also generate information with global standards, under which the cartography presented in this research was elaborated. From this mapping it is observed that the values obtained for the RMSE and $R^2$ index (Fig. 9) for the range of 0 to 5 cm were 1.78% and 0.56 respectively. From 5 to

15 were 1.77% and 0.53 and from 15 to 30, 1.79% and 0.83, very similar results in the first two intervals, partly due to the dimensionality and proximity between them. Taking as reference the $R^2$ values, all higher than 0.5, this work presents better results than similar works that used the same method for SOC estimation, for example, those reported by Y. Zhang (2020), using 122 samples in an area of 7692 km$^2$, those of Xin et al. (2016) with 180 samples in 72 km$^2$ or those of Yao (2019) using 90 samples, which obtained $R^2$ values of 0.21, 0.2 and 0.4 respectively.

A factor not considered in this work was the validation of the attributes of the horizon properties in a simple or combined way to identify outliers, for example, using Tukey's rule (Pham et al., 2019) or out of range (pH values less than 0 or greater than 14). This omission was due to the fact that a large part of the horizons did not have assigned values. As shown in Table 3, only three attributes (pH, clay and sand) exceed 70% of records with values, while another two (silt and organic carbon) have just over 50% values. The other attributes do not exceed 40%, there are even three properties with less than 6%, which

are inorganic carbon, coarse fragments and water retention. The above was a factor that influenced the choice of the area for the case study, it is important to have data, but also that they are complete.

A possible reason why the profiles have been provided incomplete may be the one mentioned by Arrouays et al. (2017) or Rossiter (2004), about privacy or data ownership policies, in addition to institutional, legal and cultural factors, prevent data from being fully shared. Breaking down those barriers would allow that data to be used by a larger number of global users.

Given the importance of these databases, it is pertinent to make new efforts to collect data from other sources, such as research centers or universities, in order to strengthen this or other databases. As shown in the analysis of SOC variability, this revised version of SISLAC data offers the potential to generate information that helps decision-making on issues in which soils are decisive. It can also be used to plan future soil surveys in areas with low density or where updated information is required. Another possible use of these data may be to improve existing information (in scale and depth), such

as the Organic Carbon Map (FAO & ITPS, 2018), or to generate new information such as that presented by Gutierrez (2020) using SISLAC data.

In summary, from the initial data set, 15% of profiles were excluded and another 4.5% were corrected. This work tried to exclude as few profiles as possible given their importance in areas with low spatial density. Furthermore, as mentioned by Hengl (2019), this data is the only thing available at this time in many places, so its availability is important. Knowing the

level of integrity of the data, what the main problems are and where they occur, can help the countries involved to know where to put more efforts to have more reliable data. In that sense, this work may contribute to support soil conservation efforts, increase food and water security, maintain healthy ecosystems, and reduce climate change's impact.

## 5 Data availability

The data is available at https://doi.org/10.5281/zenodo.6540710 (Díaz-Guadarrama, S. & Guevara, M., 2022) in three
different formats: Comma-Separated Values (.csv), Microsoft Access Database (.mdb), and as PostgreSQL – PostGIS
Database backup. The source code used is located at the same repository.

## 6 Conclusions

This work was successful in improving the SISLAC database, thus generating a revised database version in which all the soil
profiles have high quality and completeness to be efficiently used in multiple applications (e.g., digital soil carbon mapping
and reporting). In the revised SISLAC database, 15% of soil profiles were excluded (e.g., horizon information duplicated or
overlapped) and 4.5% of the soil profiles were adjusted to the same data structure. We demonstrate the usability of the
revised SISLAC database developing a reproducible digital soil carbon mapping framework which improves the analysis of
soil carbon and depth relationships from a discrete to a continuous fashion. In our usability example we observe relatively
high accuracy ($R^2$ of  0.5 and RMSE 1.78), demonstrating the potential of databases such as SISLAC to generate information
on the spatial variability of soils across large areas with high spatial detail. The database used is a product of the cooperation
of national institutions of the countries of the region, investing efforts in the collection of additional data, for example, those
produced in universities or research centers could lead to an increase in the volume of the revised version of SISLAC (as
new and better data become available), and these in turn, may allow the generation of new spatial information on soil
properties to improve what is currently available.

## Competing interests

The authors declare that they have no conflict of interest.

## Acknowledgements

Sergio Díaz acknowledges support by the Colombian Institute of Educational Credit and Technical Studies Abroad –
ICETEX.
Mario Guevara acknowledges support from grants: UNESCO-IGCP-IUGS, 2022 (#765), UNAM-PAPIIT, 2021
(#IA204522) and USDA-NIFA-AFRI, USA, 2019 (#2019-67022-29696).
We would also like to thank the people who contributed data to SISLAC from their institutions: Bolivia: Miguel Ángel
Vaca; Chile: José Sergei Padarian Campusano; Colombia: Oscar Daniel Beltrán Rodríguez, Napoleón Ordoñez Delgado,
Javier Otero García, Rafael Antonio Pedraza Rute and Reinaldo Sánchez López; Costa Rica: Bryan Alemán Montes;
Ecuador: María Natalia Rumazo Chiriboga and Darwin Sánchez Rodríguez; El Salvador: Edgard Mayen; Guatemala: Juan





Antonio Padilla Cruz and Claudia Cecilia Saput; Honduras: Arturo Varela Ocón; Nicaragua: Jose Ariel Cruz Martínez and Wilmer Rodríguez; Perú: Germán Belizario-Quispe, Marcos Gabriel Cerna Arellano, Alberto Cortez Farfán, José Carlos De la Cruz Espinoza, Gouri Agusto Aparicio Cavero, Gabriel Máximo Larota Cantuta, Efraín Oscar Rosario Sánchez, Kharolyn Elizabeth Santander Hidalgo Candia, Raúl Uscamayta Quispe and Jorge Vásquez Acuña; Uruguay: Inés Barilani, Gastón
Bentancor, Gonzalo Daniel Pereira Facal and Claudio Prieto.

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
