# Peer review of "Improving Latin American Soil Information Database for Digital Soil Mapping enhances its usability and scalability."

_Earth System Science Data, 2022_

## Author Comment (AC1)

**General comments from RC1**

The manuscript "Improving Latin American Soil Information Database for Digital Soil Mapping enhances its usability and scalability" submitted to ESSD described a method to identify the main problems in the SISLAC profiles occur systematically in Latin American countries, and provided a work flow to identify the errors in SISLAC, and finally, the authors carefully checked the errors in the SISLAC database and provided a quality improved SISLAC. This work shows the potential of improved soil databases for the generation of spatial information such as SOC or any other property which have been surveyed in existing regional or national scale soil datasets, and it has the potential to improve the global scale soil datasets. I only have few minor suggestions for the authors to consider and to correct. Other than that, I believe this work contributed to improve the quality of an existing soil dataset and their works is important in data science community.

 Some minor suggestions:

1. **Line 160: how about the sites coincided with their respective country, but may have other issues?**

Response: We recognize that there could be other issues with the sample location. However, we are working with a database of a continental region; at this level, the only spatial check that we can perform is the one of belonging.  There is no way to check that they are not correctly located because, in the original database, we just have the "country" as a variable; we do not have another specific variable.

2. **Line 162: Figure 3c is an example of coordinates inverted, but why it was marked as correct in the figure (marked as √)?**

Response: (**Line 162**) the symbol in the image has been changed, the circle is marked with "!" indicating alert. In the paragraph, it is clarified that they were checked by inverting coordinates, and when these profiles coincided they were validated.

3. **Line 174: can you explain when and why gaps exist?**

Response: (**Line 174**) Gaps can occur for reasons such as: the data was not taken at the site, loss of data in the office, or error or omission in transcription.

4. **Line 314: "This work is a effort" should be "This work is an effort".**

Response: This part was removed.

5. **Line 314-324: this paragraph talked about improving SISLAC contribute to a better data in the region (national results such as Colombia, Ecuador, and Argentina), how about its contribution to the global soil dataset? Are SISLAC be included in the global soil datasets such as SoilGrid, SoilGrid2, HWSD? How and whether the approach used in this study can be applied to improve global soil datasets?**

Response: The new version is free to use without restrictions (we provided a DOI for downloading the dataset), and one of the motivations for this work is that it can be used by as many users as possible, including those who can integrate it into global initiatives such as WoSIS or GloSIS. These additions are not within the scope of this work at the moment.

**6. Line 322: "Y. Zhang (2020)" should be "Zhang (2020)", check this issue for the entire manuscript, please.**

Response: There are two authors with the same surname and the same year (Y. Zhang (2020) and Z. Zhang (2020)), so it is specified with the initial of the first name. Additionally, the list of references was reviewed.

**7. Discussion: I suggest that subtitles can be added to increase the readability of the discussion.**

Response. Subtitles have been added.

**8. Captions of some tables and figures are too simple, and the necessary descriptions should be added to make the tables and figures self-explanatory.**

Response: The necessary descriptions were complemented.

**9. Table 1: it has a period sign (.) at the end of the table caption, but table 2 does not has one, same issue for figures, please check all figure and table captions.**

Response: The titles of tables and figures have been standardized: *Table (or Fig.) N: Description.*

**10. Table 2: PDDL, ODC-By, ODC-ODbL, CC-BY, CC-BY-NC, CC-BY-NC-ND; those are all acronyms, they should be explained.**

Response: Acronyms were explained on the table.

**11. Table 4: can you also give an example of gaps between layers exist?**

Response: For this table, as there is no error, the gaps are not included in the demonstration.

**12. Table 5: "Assign the value of the upper limit of the last layer plus 10", need to explain why "plus 10".**

Response: This rule was defined by expert judgement. Complemented in the table (Now is Table 7)

**13. Table 6: for the first case (Organic layer), I see no difference between "Inconsistency" and "Correction Guideline". Should the top be "-5" in the correction guideline column? (i.e., organic layer should be -5 to 0).**

Response: (**Table 8, line 190**) It was a mistake when pasting the image, sorry. The image has been changed to the correct one.

**Figure 3: in the brackets, panel a, b, and c were explained, why there is no description about panel d? Panel c was an example of coordinates inverted, why labeled as √ ?**

Response: (**Line 160**) The descriptions are specified in the paragraph above the table and the symbol was changed in panel 3.

14. **Figure 8: this figure looks not correct, should y axis "Residual" rather than "Predicted values"? And what are dashed lines and solid lines? They should be explained in the figure caption. Why the solid line is necessary in this figure?**

This comment corresponds to the excluded part and is therefore omitted.

---

## Author Comment (AC2)

**General comments from RC2**

The paper "Improving Latin American Soil Information Database for Digital Soil Mapping enhances its usability and scalability" describes the effort of gathering and harmonizing Latin America soil data from historical surveys, which was promoted by FAO's South American Soil Partnership and involved several collaborators across from region. The authors presented a quality assessment analysis, described a new improved version of the dataset, and demonstrated the potential of SISLAC for generating new soil information through digital soil mapping. This type of work is important in order to document soil data integration efforts and document the best practices for harmonizing heterogeneous soil datasets. In addition, it makes clear that avoiding removing a lot of data that can be simply adjusted has an enormous impact on the final number of samples and potentially the spatial representation across a region. Overall, the authors did a great job in describing their quality analysis, but I was not convinced by the results from digital soil mapping. I think the authors could rather explore the dataset with a denser descriptive analysis, avoiding a predictive approach (which was very simple and suboptimal). Therefore, I don't have any major objection to its publication. However, I think that a moderate revision of the second goal is required before reaching a final decision. Finally, I congratulate the authors for making available the improved SISLAC dataset on a public persistent repository (Zenodo) with an open-access license.

Specific comments:

1. **Although the first introduction paragraphs describe what soil is and how they form, the current structure seems a bit overloaded to me. For example, the first three sentences have a lot of information that is hard to grasp at first moment. I would suggest starting from line 72 and relocating those first sentences after explaining the soil importance, bringing the definitions after a gentler introduction.**

Response: We believe that we should leave it as it is, since it first explains what soil is, then specifies that it is composed of horizons and finally (from line 72 ) its relevance.

2. **The data are well described. I was able to access their online website (http://54.229.242.119/sislac/es) and check some soil profiles. However, I had some issues with signing up to the portal (could not confirm my email address to log in). The public access does not have any download button, but it seems the user can copy and paste single profile tabular data. They do not mention any application programming interface (API) in this data section, which is a characteristic of modern web 2.0 platforms (https://en.wikipedia.org/wiki/Web_2.0). I would suggest at least discussing data distribution through APIs and explaining in the manuscript if this feature is planned as a potential improvement of future SISLAC versions.**

Response: SISLAC is a system independent of this work; its administration and use are beyond the scope of the authors of this work, in which we focus on downloading, analyzing the data, and showing an example of its usability. As a result of our work, we offer a corrected database that is available at (DOI). It can be downloaded in .csv format as well as the script in which we made the corrections; it is specified in the Data Availability section.

3. **It is not clear in the manuscript if the SISLAC from their website is the older or the improved version.**

Response: Continuing with the previous point, the original database is the one shown on the SISLAC page, and the revised version is the one available in the repository. We do not have the possibility to change the data in SISLAC.

4. **When navigating their website, I found that many samples come from the WoSIS snapshot of 2016. There are other datasets, such as the SISINTA. I just wonder if the authors could provide an overview of the original sources (WoSIS, SISINTA, etc.) similarly to what they did with country numbers. This new table could be placed as supplementary material to help readers quickly evaluate the difference between SISLAC and other available public datasets, such as WoSIS.**

Response: (Line 246) Tables 14 and 15 have been added. The first one lists the data sources and the country to which they belong. The second indicates the percentage of soil property attributes with data in the initial database and the percentage in the new version.

5. **How do the authors expect to update SISLAC when newer versions of the original sources are released? Have they automated the quality analysis keeping in mind new updates or has this current work involved a workforce for manual inspection?**

Response: Based on the current work, we intend to follow up with new updates to the database with the integrity controls implemented so that the data meet minimum integrity requirements. As indicated (Line 94) these efforts should be led by FAO as coordinator of the region to update this database.

6. **Why the authors defined 150 cm as the bottom limit instead of 200 cm? 200 cm is an arbitrary convention from pedology but at least is the standard limit of GlobalSoilMap. A simple justification would be enough in my view, as reprocessing the data would be very expensive.**

Response: This question corresponds to the excluded part; however, this measure was taken since it was the common limit in the selected area.

7. **Both good-of-fitness equations have minor mistakes, although the result will not be impacted as the difference between observed and predicted are squared. However, the sum of squared residuals should be observed-predicted in both RMSE and R2 numerator.**

Response: This question corresponds to the excluded part.

8. **The authors did a good job of describing and reporting their quality assessment analysis. I wonder if they used some published guidelines or proposed those based on the issues they faced in the project development. I think this data description paper and methods can help many other efforts for soil data integration and harmonization.**

Response: As indicated in line 262, the criteria for the analysis of the databases consisted of an expert judgment of the region, and the rules applied coincide with those implemented in the work of Batjes (1995) and Leenaars (2013).

9. **I only have serious concerns about the results from the data usability section. The authors provided reasonable summary statistics and visualizations. However, the cross-validation statistics are very intriguing, at least from the current scatterplot visualization. In my view, it is impossible to get moderate to good R2 from the scatter distribution they plotted, especially for the third panel where**

**they reached an R2 of 0.83. All the fitted lines are almost flat, with a narrower predicted variance compared to the original values. In addition, when many data points are overlapped, it is common to present a scatterplot with point density, making possible the evaluation of the linear trend around the fitted line. The bias of these models is really high, so other performance metrics like Lin's correlation concordance coefficient (CCC) would indicate a potential unsatisfactory performance. Therefore, I'm not convinced with the results from this data usability section and even question the authors if they are willing to keep these results in their manuscript. Instead of presenting these questionable results from digital soil mapping or another predictive approach, I think the authors could rather crunch the dataset with a denser exploratory data analysis with summary statistics, multivariate data analysis using PCA in combination with grouping factors (coloring by color, biome, or any other physical information), some spatial statistics (like Moran's index, or even screening variograms for the whole region), etc. In my opinion, those results would be a greater fit for the manuscript type, which is a data description paper. If they follow this suggestion, I think they should adjust the paper title.**

Response: We agree with the comments, the part of the digital soil mapping has deficiencies and takes us away from the central objective of the article which is the database. For these reasons it was decided to exclude this part in the new version of the manuscript.

**10. The discussion is well developed; however, I would only suggest adjusting it if the digital soil mapping results are revised.**

Response: This question corresponds to the excluded part.

**Technical corrections:**

**Overall, the paper is clear and well-structured. I'm not an English native speaker, but I think the readers would benefit from a proofread version of the paper.**

**In line 214, I think the authors should define ordinary kriging as an interpolation method rather than a method to estimate SOC, e.g.: "On the other hand, ordinary kriging (OK) was used for horizontal variability assessment, a method frequently used to spatially predict SOC …"**

Response: This question corresponds to the excluded part.

---

## Author Comment (AC3)

**Dear revisor (RC1):**

We greatly appreciate the time you have devoted to our manuscript. Below we respond to the suggestions you indicated, which served to improve our manuscript. There are two important changes in the document; the first is that in a joint effort with FAO´s Latin America and the Caribbean Soil Partnership, a review was made of the databases available in the region and we were able to consolidate a larger database, which has grown from 41,000 records to almost 67,000. This is reflected in the new manuscript along with the new DOI of the dataset, which are made available to the soil science scientific community under the FAIR (Findable, Accessible, Interoperable and Reusable) principles. The second is that at the suggestion of another reviewer, the digital soil mapping part has been excluded in order to focus on the description of the SISLAC database and the methodology used for its analysis. Once again, we thank you for your time and comments that have enriched this work.

**General comments from RC1**

The manuscript "Improving Latin American Soil Information Database for Digital Soil Mapping enhances its usability and scalability" submitted to ESSD described a method to identify the main problems in the SISLAC profiles occur systematically in Latin American countries, and provided a work flow to identify the errors in SISLAC, and finally, the authors carefully checked the errors in the SISLAC database and provided a quality improved SISLAC. This work shows the potential of improved soil databases for the generation of spatial information such as SOC or any other property which have been surveyed in existing regional or national scale soil datasets, and it has the potential to improve the global scale soil datasets. I only have few minor suggestions for the authors to consider and to correct. Other than that, I believe this work contributed to improve the quality of an existing soil dataset and their works is important in data science community.

 Some minor suggestions:

1.  **Line 160: how about the sites coincided with their respective country, but may have other issues?**

Response: We recognize that there could be other issues with the sample location. However, we are working with a database of a continental region; at this level, the only spatial check that we can perform is the one of belonging.  There is no way to check that they are not correctly located because, in the original database, we just have the "country" as a variable; we do not have another specific variable.

2.  **Line 162: Figure 3c is an example of coordinates inverted, but why it was marked as correct in the figure (marked as √)?**

Response: (**Line 162**) the symbol in the image has been changed, the circle is marked with "!" indicating alert. In the paragraph, it is clarified that they were checked by inverting coordinates, and when these profiles coincided, they were validated.

3.  **Line 174: can you explain when and why gaps exist?**

Response: this part was explained in Line 173 Gaps can occur for reasons such as: the data was not taken at the site, loss of data in the office, or error or omission in transcription.

**4. Line 314: "This work is a effort" should be "This work is an effort".**

Response: This part was removed.

**5. Line 314-324: this paragraph talked about improving SISLAC contribute to a better data in the region (national results such as Colombia, Ecuador, and Argentina), how about its contribution to the global soil dataset? Are SISLAC be included in the global soil datasets such as SoilGrid, SoilGrid2, HWSD? How and whether the approach used in this study can be applied to improve global soil datasets?**

Response: The new version is free to use without restrictions (we provided a DOI for downloading the dataset), and one of the motivations for this work is that it can be used by as many users as possible, including those who can integrate it into global initiatives such as WoSIS or GloSIS. These additions are not within the scope of this work at the moment.

**6. Line 322: "Y. Zhang (2020)" should be "Zhang (2020)", check this issue for the entire manuscript, please.**

Response: There are two authors with the same surname and the same year (Y. Zhang (2020) and Z. Zhang (2020)), so it is specified with the initial of the first name. Additionally, the list of references was reviewed.

**7. Discussion: I suggest that subtitles can be added to increase the readability of the discussion.**

Response. Subtitles have been added.

**8. Captions of some tables and figures are too simple, and the necessary descriptions should be added to make the tables and figures self-explanatory.**

Response: The necessary descriptions were complemented.

**9. Table 1: it has a period sign (.) at the end of the table caption, but table 2 does not has one, same issue for figures, please check all figure and table captions.**

Response: The titles of tables and figures have been standardized: *Table (or Fig.) N: Description.*

**10. Table 2: PDDL, ODC-By, ODC-ODbL, CC-BY, CC-BY-NC, CC-BY-NC-ND; those are all acronyms, they should be explained.**

Response: Acronyms were explained on the table.

**11. Table 4: can you also give an example of gaps between layers exist?**

Response: For this table, as there is no error, the gaps are not included in the demonstration.

**12. Table 5: "Assign the value of the upper limit of the last layer plus 10", need to explain why "plus 10".**

Response: This rule was defined by expert judgement. Complemented in the table (Now is Table 7)

13. **Table 6: for the first case (Organic layer), I see no difference between "Inconsistency" and "Correction Guideline". Should the top be "-5" in the correction guideline column? (i.e., organic layer should be -5 to 0).**

Response: (**Table 8, line 190**) We apologize for a mistake when pasting the image. The image has been changed to the correct one.

**Figure 3: in the brackets, panel a, b, and c were explained, why there is no description about panel d? Panel c was an example of coordinates inverted, why labeled as √ ?**

Response: (**Line 160**) The descriptions are specified in the paragraph above the table and the symbol was changed in panel 3.

14. **Figure 8: this figure looks not correct, should y axis "Residual" rather than "Predicted values"? And what are dashed lines and solid lines? They should be explained in the figure caption. Why the solid line is necessary in this figure?**

This comment corresponds to the excluded part and is therefore omitted.

---

## Author Comment (AC4)

**Dear revisor (RC2)**

We appreciate your time in reviewing our manuscript. After your suggestions, we have decided to exclude the part of the digital soil mapping to focus on the database and its description through a principal component analysis. In addition, after working together with FAO´s Latin America and the Caribbean Soil Partnership in the last few months, we were able to consolidate a larger database, which has grown from 41 thousand records to a little more than 66 thousand after the revision and incorporation of other soil databases available in the region. This is reflected in the new manuscript and the new DOI of the dataset (https://doi.org/10.5281/zenodo.787673). This is undoubtedly great news for the soil science community in the region. It is hoped that this database will continue to grow with other similar efforts and that digital cartographic products supported by these data can be generated. Once again, we thank you for your time and send the response to your comments.

**General comments from RC2**

The paper "Improving Latin American Soil Information Database for Digital Soil Mapping enhances its usability and scalability" describes the effort of gathering and harmonizing Latin America soil data from historical surveys, which was promoted by FAO's South American Soil Partnership and involved several collaborators across from region. The authors presented a quality assessment analysis, described a new improved version of the dataset, and demonstrated the potential of SISLAC for generating new soil information through digital soil mapping. This type of work is important in order to document soil data integration efforts and document the best practices for harmonizing heterogeneous soil datasets. In addition, it makes clear that avoiding removing a lot of data that can be simply adjusted has an enormous impact on the final number of samples and potentially the spatial representation across a region. Overall, the authors did a great job in describing their quality analysis, but I was not convinced by the results from digital soil mapping. I think the authors could rather explore the dataset with a denser descriptive analysis, avoiding a predictive approach (which was very simple and suboptimal). Therefore, I don't have any major objection to its publication. However, I think that a moderate revision of the second goal is required before reaching a final decision. Finally, I congratulate the authors for making available the improved SISLAC dataset on a public persistent repository (Zenodo) with an open-access license.

Specific comments:

1. **Although the first introduction paragraphs describe what soil is and how they form, the current structure seems a bit overloaded to me. For example, the first three sentences have a lot of information that is hard to grasp at first moment. I would suggest starting from line 72 and relocating those first sentences after explaining the soil importance, bringing the definitions after a gentler introduction.**

Response: We believe that we should leave it as it is, since it first explains what soil is, then specifies that it is composed of horizons and finally (from line 72 ) its relevance.

2. **The data are well described. I was able to access their online website (http://54.229.242.119/sislac/es) and check some soil profiles. However, I had some issues with signing up to the portal (could not confirm my email address to log in). The public access does not have any download button, but it seems the user can copy and paste single profile tabular data. They do not mention any application programming interface (API) in this data section, which is a**

**characteristic of modern web 2.0 platforms (https://en.wikipedia.org/wiki/Web_2.0). I would suggest at least discussing data distribution through APIs and explaining in the manuscript if this feature is planned as a potential improvement of future SISLAC versions.**

Response: SISLAC is a system independent of this work; its administration and use are beyond the scope of the authors of this work, in which we focus on downloading and analyzing the data and showing an example of its usability. As a result of our work, we offer a corrected database available at https://doi.org/10.5281/zenodo.7876731. It can be downloaded in .csv format and the script in which we made the corrections is specified in the Data Availability section.

**3. It is not clear in the manuscript if the SISLAC from their website is the older or the improved version.**

Response: Continuing with the previous point, the original database is the one shown on the SISLAC page, and the revised version is the one available in the repository. We cannot change the data in SISLAC.

**4. When navigating their website, I found that many samples come from the WoSIS snapshot of 2016. There are other datasets, such as the SISINTA. I just wonder if the authors could provide an overview of the original sources (WoSIS, SISINTA, etc.) similarly to what they did with country numbers. This new table could be placed as supplementary material to help readers quickly evaluate the difference between SISLAC and other available public datasets, such as WoSIS.**

Response: (Line 255) Tables 14 and 15 have been added. The first one lists the data sources and the country to which they belong. The second indicates the percentage of soil property attributes with data in the initial database and the percentage in the new version.

**5. How do the authors expect to update SISLAC when newer versions of the original sources are released? Have they automated the quality analysis keeping in mind new updates or has this current work involved a workforce for manual inspection?**

Response: Based on the current work, we intend to follow up with new updates to the database with the integrity controls implemented so that the data meet minimum integrity requirements. As indicated (Line 93) these efforts should be led by FAO as coordinator of the region to update this database.

**6. Why the authors defined 150 cm as the bottom limit instead of 200 cm? 200 cm is an arbitrary convention from pedology but at least is the standard limit of GlobalSoilMap. A simple justification would be enough in my view, as reprocessing the data would be very expensive.**

Response: This question corresponds to the excluded part; however, this measure was taken since it was the common limit in the selected area.

**7. Both good-of-fitness equations have minor mistakes, although the result will not be impacted as the difference between observed and predicted are squared. However, the sum of squared residuals should be observed-predicted in both RMSE and R2 numerator.**

Response: This question corresponds to the excluded part.

**8. The authors did a good job of describing and reporting their quality assessment analysis. I wonder if they used some published guidelines or proposed those based on the issues they faced in the project**

**development. I think this data description paper and methods can help many other efforts for soil data integration and harmonization.**

Response: As indicated in line 312, the criteria for analyzing the databases consisted of an expert judgment of the region, and the rules applied coincide with those implemented in the work of Batjes (1995) and Leenaars (2013).

9. **I only have serious concerns about the results from the data usability section. The authors provided reasonable summary statistics and visualizations. However, the cross-validation statistics are very intriguing, at least from the current scatterplot visualization. In my view, it is impossible to get moderate to good R2 from the scatter distribution they plotted, especially for the third panel where they reached an R2 of 0.83. All the fitted lines are almost flat, with a narrower predicted variance compared to the original values. In addition, when many data points are overlapped, it is common to present a scatterplot with point density, making possible the evaluation of the linear trend around the fitted line. The bias of these models is really high, so other performance metrics like Lin's correlation concordance coefficient (CCC) would indicate a potential unsatisfactory performance. Therefore, I'm not convinced with the results from this data usability section and even question the authors if they are willing to keep these results in their manuscript. Instead of presenting these questionable results from digital soil mapping or another predictive approach, I think the authors could rather crunch the dataset with a denser exploratory data analysis with summary statistics, multivariate data analysis using PCA in combination with grouping factors (coloring by color, biome, or any other physical information), some spatial statistics (like Moran's index, or even screening variograms for the whole region), etc. In my opinion, those results would be a greater fit for the manuscript type, which is a data description paper. If they follow this suggestion, I think they should adjust the paper title.**

Response: We agree with the comments. The part of the digital soil mapping has deficiencies and takes us away from the central objective of the article, which is the database. For these reasons, it was decided to exclude this part in the new version of the manuscript and focus on the database description through a principal component analysis (ACP). The ACP included five soil variables (those with the major number of records in the database), two profile attributes (number of horizons and profile depth), and a categorical variable (soil group – based on the WRB taxonomic system). We think that this correction help us and our readers to understand the potential of the use of this new database.

10. **The discussion is well developed; however, I would only suggest adjusting it if the digital soil mapping results are revised.**

Response: This question corresponds to the excluded part.

**Technical corrections: Overall, the paper is clear and well-structured. I'm not an English native speaker, but I think the readers would benefit from a proofread version of the paper.**

**In line 214, I think the authors should define ordinary kriging as an interpolation method rather than a method to estimate SOC, e.g.: "On the other hand, ordinary kriging (OK) was used for horizontal variability assessment, a method frequently used to spatially predict SOC ..."**

Response: This question corresponds to the excluded part.

---

## Author Response (AR2)

Dear Editors,

"We appreciate the acceptance of our manuscript. Below, we describe the adjustments made:"

- Line 76: a space is needed before (FAO, 2017). This adjustment has been made.
- Figure 2: "1-Data validation processes" change to "Data validation processes". This adjustment has been made.

"Thank you once again for considering our manuscript. We appreciate your time and attention to our work. We look forward to any further feedback or guidance you may provide.

Sincerely,

Sergio Díaz-Guadarrama & SISLAC TEAM